# Preferences regarding COVID-19 vaccination among 12,000 adults in China: A cross-sectional discrete choice experiment

Fengyun Yu[1‡], Lirui Jiao[2‡], Qiushi Chen[3‡], Qun Wang[4], Manuela De Allegri[5], Zhong Cao[6], Wenjin Chen[7], Xuedi Ma[7], Chao Wang[7], Jonas Wachinger[5], Zhangfeng Jin[8], Aditi Bunker[5], Pascal Geldsetzer[9,10], Juntao Yang[7]*, Lan Xue[11,12]*, Till Bärnighausen[5,7‡], Simiao Chen[5,7‡]*

1 Interdisciplinary Centre for Scientific Computing, University of Heidelberg, Heidelberg, Germany, 2 Department of Health Policy and Management, Gillings School of Global Public Health, University of North Carolina at Chapel Hill, Chapel Hill, North Carolina, United States of America, 3 The Harold and Inge Marcus Department of Industrial and Manufacturing Engineering, The Pennsylvania State University, University Park, Pennsylvania, United States of America, 4 Faculty of Humanities and Social Sciences, Dalian University of Technology, Dalian, China, 5 Heidelberg Institute of Global Health (HIGH), Faculty of Medicine and University Hospital, Heidelberg University, Heidelberg, Germany, 6 State Key Lab of Intelligent Technologies and Systems, Department of Automation, Tsinghua University, Beijing, China, 7 School of Population Medicine and Public Health, Chinese Academy of Medical Sciences & Peking Union Medical College, Beijing, China, 8 School of Economics, Zhejiang University of Technology, Hangzhou, China, 9 Department of Medicine, Division of Primary Care and Population Health, Stanford University School of Medicine, Stanford, California, United States of America, 10 Chan Zuckerberg Biohub – San Francisco, San Francisco, California, United States of America, 11 Institute for AI International Governance, Tsinghua University, Beijing, China, 12 School of Public Policy and Management, Tsinghua University, Beijing, China

‡ FY, LJ and QC contributed equally to this study as co-first authors. TB and SC are co-senior authors to this work.
* yangjt@pumc.edu.cn (JY); xuelan@mail.tsinghua.edu.cn (LX); simiao.chen@uni-heidelberg.de (SC)

**Data Availability Statement:** All data underlying the findings of this study are publicly available and can be accessed from the GitHub repository:

## Abstract

Understanding public preferences concerning vaccination is critical to inform pandemic response strategies. To investigate Chinese adults' preferences regarding COVID-19 vaccine attributes, we conducted a cross-sectional online survey in 12,000 Chinese adults in June-July, 2021. Participants were requested to answer a series of discrete choice questions related to hypothetical COVID-19 vaccines. Using mixed logit models, our analysis revealed that participants had a higher preference for COVID-19 vaccines with longer duration of protection (coefficient: 1.272, 95% confidence interval [1.016 to 1.529]) and higher efficacy (coefficient: 1.063, [0.840, 1.287]). Conversely, participants demonstrated a lower preference associated with higher risk of rare but serious side-effects (coefficient: -1.158, [-1.359, -0.958]), oral administration (coefficient: -0.211, [-0.377, -0.046]), more doses (coefficient: -0.148, [-0.296, 0.000]) and imported origin (coefficient: -0.653, [-0.864, -0.443]). Moreover, preferences were heterogeneous by individual factors: highly educated participants were more sensitive to the negative vaccine attributes including price (coefficient -0.312, [-0.370, -0.253]) and imported vaccine (coefficient -0.941, [-1.186, -0.697]); there was also substantial heterogeneity in vaccine preferences with respect to age group, marital status, work status, income, chronic diagnosis history, COVID-19 vaccination history and geographic regions. As the first study of examining the public preferences for COVID-19

https://github.com/Finley-Maple/Vaccine-DCE-China.git.

**Funding:** This study was supported by the Chinesisch-Deutsches Zentrum für Wissenschaftsförderung under grant number C0048 to SC. The funders had no role in study design, data collection and analysis, decision to publish, or preparation of the manuscript.

**Competing interests:** The authors have declared that no competing interests exists.

vaccine in China with a large nationwide sample of 12,000 adults, our results indicate that future vaccine should pose lower risk, possess longer protection period, have higher efficacy, be domestically produced, and have lower costs to increase the COVID-19 vaccination coverage. Our current study findings from this study provide insights and recommendations for not only COVID-19 vaccine design but also vaccine attribute preferences to increase vaccine uptake in potential future pandemics.

## Introduction

The coronavirus disease 2019 (COVID-19) pandemic has substantially harmed population health and economies worldwide [1–4]. Researchers, policymakers, and other stakeholders have sought effective measures to control the pandemic [5, 6]. Of the potential options, safe and effective vaccination was considered as a preferable pathway to reduce morbidity and mortality against COVID-19 pandemic [7–10]. In February 2023, thirteen COVID-19 vaccines have been approved for emergency use by World Health Organization (WHO), including inactivated vaccines, viral vector vaccines, mRNA vaccines, and protein subunit vaccines [11]. Beyond these approved vaccines, 183 vaccine candidates have been approved for clinical trials, and 199 vaccines are in preclinical trials, according to WHO data released on March 30st, 2023 [12]. However, vaccine availability alone does not necessarily lead to high vaccination rates [13]. Additionally, as COVID-19 vaccines may be regularly offered similar to flu shots in years to come [14], it is of critical importance to understand vaccine preferences and factors that affect vaccine decision-making in order to increase global vaccination coverage for both the ongoing COVID-19 responses and for future pandemic preparedness.

Discrete choice experiments (DCEs) have been frequently utilized in previous studies examining COVID-19 vaccine preferences, due to their robust and accurate estimation of health behaviors and preferences [15]. In vaccine-related DCEs, participants for example have been asked to choose between vaccines with different attributes, such as efficacy, availability, doses, country of production, price, duration of protection, and risk of side effects [15–20]. Some experiments also offered "opt-out" options for cases when the participants felt that the vaccines offered were unacceptable for them [21, 22]. Via DCEs, researchers have studied how attributes influence people's preferences and affect the uptake of a COVID-19 vaccine, but available studies commonly have focused on specific populations, such as healthcare workers [23], university students [24], or people with chronic immune-mediated inflammatory diseases [25]. Existing literature indicates that attributes including high vaccine efficacy, low risk of side effects, long duration of immunity, and low number of required doses have the highest influence on COVID-19 vaccine uptake probability. For example, a study exploring COVID-19 vaccine preferences in China found that vaccine uptake probability increased by 45.7% if vaccine effectiveness increased from 40% to 85%, and increased by 43.2% if the risk of severe side effects fell from 50/100,000 to 1/100,000 [16].

Current literature related to COVID-19 vaccine preferences has yet to utilize nationwide samples that contain more than a few thousand participants to study the vaccine preferences of the general population. With the relatively small samples in these studies, DCEs have limited power to identify heterogeneities and interactions between vaccine attributes and individual characteristics with further details. Furthermore, limited research has closely examined the geographical heterogeneities with regards to vaccine uptake probability. To fill these gaps, this study investigated vaccine preferences and associated determinants for COVID-19 vaccines

among individuals in China utilizing a large, nationwide sample. In addition, we predicted the changes in vaccine uptake probability when varying one attribute from the baseline vaccine to provide insights into the impact of vaccine attributes on vaccine uptake. Furthermore, we predicted the vaccine uptake probability of hypothetical Sinovac-CoronaVac (COVID-19) vaccines at different prices for different provinces, to demonstrate pricing-dependent geographical heterogeneities in vaccine uptake. This study aimed to facilitate the development of vaccination strategies, such as targeted vaccine campaigns, and provided empirical findings that can elucidate optimal vaccine pricing for anticipated vaccination rates across subpopulations for relevant policymaking. Through this research, we hope to inform best practices in promoting future vaccine uptake.

## Methods

### Survey development, attributes, and attribute levels

We developed the survey, "Questionnaire on Willingness to COVID-19 Vaccination," which included questions in four sections: choice tasks, participants' basic information, health conditions, and historical vaccination behaviors (S1 Text). In the first section, we investigated adults' preferences for the COVID-19 vaccines using a DCE. Specifically, each participant was presented with and expected to choose between two competing scenarios representing two different hypothetical COVID-19 vaccines that contained different sets of attributes. We also provided an opt-out option because COVID-19 vaccinations were not mandatory in China at the time of our study [26].

A sequential mixed-methods approach [27] was applied to develop vaccine attributes and their corresponding levels (S2 Text), including a review of the available literature and in-depth qualitative interviews to identify suitable attributes and attribute levels. The chosen attributes of the hypothetical vaccines are listed as follows (Table 1): total price (free, 200 RMB ≈ \$30.8, 400 RMB ≈ \$61.6, 600 RMB ≈ \$92.4), risk of rare but serious side-effects such as temporary or permanent paralysis from the vaccine (no risk: 0, moderate risk: 1/1,000,000, high risk: 1/100,000), duration of protection (6 months, 12 months, life-long), degree of efficacy (50%, 70%, 90%), method of vaccine administration (injection, oral), frequency of vaccination (one dose, two doses, three doses), and vaccine origin (domestic, imported). These seven attributes resulted in 1,296 (= $4*3^4*2^2$) hypothetical vaccine profiles and a total of 839,160 possible pairwise choice scenarios. We used a sequential orthogonal design to reduce the choice scenarios to 36 choice tasks in three blocks, with 12 choice tasks in each block [27]. We assigned each participant randomly to one of the three blocks to complete the survey.

Prior to finalizing the DCE design, we recruited 45 adults into a pilot study with similar characteristics to the target population in June 2021 to validate the clarity of vaccine attribute definitions, confirm the appropriateness of size of choice sets, and generate parameters for the final design [28, 29]. Based on the parameters from the pilot study, we used the Ngene (version 1.2.1) DCE design software package to generate a series of choice tasks for participants to choose. We did not add an exercise choice set to help respondents get familiar with the choice tasks due to the following reasons. First, nearly all participants passed the internal consistency test in the pilot study and had no cognitive difficulties in understanding the choice set. Second, it takes approximately 10 minutes to finish the actual choice set, and so an additional exercise choice set would increase the time burden. See S2 Text for more details on the process of developing the DCE and S1 Table for an example of choice tasks provided to participants.

**Table 1. Discrete choice experiment attributes and attribute levels.**

| Attribute label | Labels of plausible levels | Encoding form | Prior parameters from the pilot study | Explanation and derivation |
|---|---|---|---|---|
| Total Price | • Free<br>• 200 RMB (≈$30.8)<br>• 400 RMB (≈$61.6)<br>• 600 RMB (≈$92.4) | Continuous | -0.0012 | Total price for the vaccination.<br>The levels are derived from the focused interviews. |
| Risk of rare but serious side-effects from the vaccine | • No risk: 0;<br>• Moderate risk: 1/1,000,000<br>• High risk: 1/100,000 | Categorical | • Reference<br>• 0.56<br>• -0.98 | Serious but rare side effects might largely impact participants' preferences. The levels are derived from Schwarzinger, Watson [9] |
| Duration of protection | • 6 months;<br>• 12 months;<br>• life long | Categorical | • Reference<br>• 0.44<br>• 2.23 | The levels are derived from the focused interviews. |
| Degree of efficacy | • 50%;<br>• 70%;<br>• 90% | Categorical | • Reference<br>• 1.30<br>• 1.64 | The levels are derived from the focused interviews. |
| Vaccine administration | • Injection;<br>• Oral | Categorical | • Reference<br>• -0.23 | The levels are derived from the focused interviews. |
| Frequency of vaccination | • One dose;<br>• Two doses;<br>• Three doses | Categorical | • Reference<br>• -0.46<br>• -0.48 | Total doses are needed to achieve efficacy. |
| Vaccination origin | • Domestic;<br>• Imported | Categorical | • Reference<br>• -0.15 | Vaccine patriotism might affect participants' choices. It can be validated in the pilot study. |

## Data collection

We conducted the online survey among the general adult population of China between June 4th and July 11th in 2021. The survey was distributed via KuRunData, a private online survey company that maintains a database of potential survey participants and administers surveys. KuRunData takes advantage of its own platform and partnerships with other websites in recruiting participants, and also encourages registered members to recruit new members through the popular mobile application WeChat Mini [30]. KuRunData verified that members have access to mobile phones, internet, and can navigate online surveys. In this study, KuRunData sampled between 360–480 participants in each of China's 31 provincial-level administrative units, adjusting the specific sample size with respect to each unit's entire population, with a total target sample size of 12,000 adults. Within each province, KuRun-Data sampled a proportion of participants reflective of the demographic composition of the province's population (as per the 2019 census [31]) by sex, age group, and urban-rural residence.

Adults in the survey pool could participate in the survey on KuRunData's platform until the total target sample size threshold was reached. Survey participants would receive 2–5 RMB (depending on their membership level in KuRunData platform) upon the completion of the questionnaire. Before filling out the questionnaire, participants provided their informed written consent with signature confirmation. In total, 12,193 adults agreed to take the online survey, among which 193 samples were excluded for not meeting the following requirements: 1) the participants must complete this survey between the time period of 360–1000 seconds, and 2) the age, marital status and education status should be consistent. This resulted in 12,000

total valid samples to be included in our final analysis. Further details of our data quality control methods are provided in S2 Text.

### Ethical review

This study was approved by the Ethics Review Committee for Biomedical Research Involving Humans of the Chinese Academy of Medical Sciences and Peking Union Medical College (CAMS&PUMC-IEC-2020-001) and the Ethics Committee of the Medical Faculty of Heidelberg University (S-041/2021). All participants provided online informed consent before completing the survey.

### Inclusivity in global research

Additional information regarding the ethical, cultural, and scientific considerations specific to inclusivity in global research is included in the S1 Checklist.

### Data analysis

To model an individual's vaccine preference with different attributes, we started by defining the individual's utility function based on vaccine attributes and attribute levels [32]. Following the random utility maximization model framework [33], each participant chose alternative $j$ that yields the maximum utility among the three presented alternatives [34]. We specified the utility $U_{ij}$ associated with alternative $j$ for individual $i$ in each choice scenario as:

$$U_{ij} = V_{ij} + \epsilon_{ij} \tag{1}$$

where $V_{ij}$ is the deterministic utility and $\epsilon_{ij}$ is the random utility that was assumed to follow a Type 1 extreme value distribution. In particular, the deterministic utility $V_{ij}$ was parameterized as a linear function of the seven vaccine attributes (with price coded as a continuous variable and the other categorical attributes coded as dummy variables), in addition to two alternative-specific constants accounting for the tendency to vaccinate and the tendency to choose left options over right ones, respectively [35]. As a result, the deterministic utility that individual $i$ obtained from an alternative $j$ admitted the following mathematical form:

$$\begin{aligned} V_{ij} &= ASC_j + ASC\_1st_j + \beta_{1i}Price_{ij} + \beta_{2i}Risk\_moderate_{ij} \\ &+ \beta_{3i}Risk\_high_{ij} + \beta_{4i}Duration\_middle_{ij} + \beta_{5i}Duration\_long_{ij} \\ &+ \beta_{6i}Efficacy\_70\%_{ij} + \beta_{7i}Efficacy\_90\%_{ij} + \beta_{8i}Oral_{ij} \\ &+ \beta_{9i}Dose\_2_{ij} + \beta_{10i}Dose\_3_{ij} + \beta_{11i}Imported_{ij} \end{aligned} \tag{2}$$

where $ASC_j$ is an alternative-specific constant of preference for having a vaccine (versus opt-out), $ASC\_1st_j$ is a constant accounting for the first option, $\beta_{1i}, \ldots, \beta_{11i}$ are the preference parameters of each attribute level for individual $i$. To account for preference heterogeneity among the population, we considered the preference parameters as random effects. That is, we applied a mixed logit model [34] to investigate adults' preferences for vaccines with different attributes. In our basic mixed logit model, we assumed preference parameters $\beta_{mi}$ to follow a normal distribution,

$$\beta_{mi} = \mu_m + \xi_{mi}, \; \xi_{mi} \sim N\left(0, \delta_m^2\right), \; m = 1, \ldots, 11, \tag{3}$$

where $\mu_m$ is the mean of the preference parameter and $\xi_{mi}$ captures the individual-level heterogeneity.

To further examine the heterogeneity in the preference parameter $\beta_{mi}$ in the population, we considered an extended mixed logit model by incorporating the interaction effects between individual's characteristics and the preference parameter for those vaccine attribute levels with significant heterogeneity [28]. That is, if the variance estimate $\hat{\delta}_m^2$ was shown to be significantly different from 0 based on model (3), we re-parameterized vaccine preference parameter $\beta_{mi}$ by incorporating individual participant's characteristics as:

$$\beta_{mi} = v_m + \sum_k \gamma_{mk} Z_{ik} + \zeta_{mi}, \quad \zeta_{mi} \sim N(0, \sigma_m^2), \quad m = 1, \ldots, 11, \tag{4}$$

where $v_m$ is the mean of the preference parameter for the reference population, $Z_{ik}$ represents the characteristic $k$ of participant $i$, $\gamma_{mk}$ denotes interaction effect between $m$-th vaccine attribute level and $k$-th individual characteristic, and $\xi_{mi}$ captures participant $i$'s unobserved preference heterogeneity for attribute $m$. For participants' characteristics, we included sex (male and female), age group (18–39 years, 40–59 years, $\geq$ 60 years), education (primary school or less, middle and high school, college and above), current residence (rural, urban), work situation (work in a private setting, work in a public setting, no job), insurance type (Urban Employees Basic Medical Insurance [UEBMI], Basic Medical Insurance for Urban and Rural Residents [BMIURR]), yearly household income level ($<$ 60,000 RMB, 60,000–149,999 RMB, $\geq$ 150,000 RMB), whether the participant had been diagnosed with any chronic disease, and whether they had been vaccinated with at least one dose of vaccine.

Our basic mixed logit model showed that a vaccine administered in two doses had non-significant standard deviations (SD) for its random effect (see S2 Table for detailed results). Therefore, we did not include this attribute when adding the interaction with other participants' covariates in the extended mixed logit model.

With all the model parameters estimated from the extended mixed logit model (4), we calculated the predicted vaccine uptake probability of each participant, assuming only two choices available: vaccination vs. opt-out. We used the following formula to compute the vaccine uptake probability of individual $i$ for vaccine alternative $j$:

$$P_{ij} = \frac{\exp\left(V_{ij}\right)}{\exp\left(V_{ij}\right) + 1}, \tag{5}$$

where utility $V_{ij}$ was obtained from Eq (2) and the preference parameter $\beta_{mi}$ for calculating the utility $V_{ij}$ was estimated using model (4).

Lastly, to examine the geographical heterogeneity in adults' preferences for vaccines, we stratified our analysis by province using the basic mixed logit model (3) (considering that the sample size in each province may not be sufficiently large for applying the extended mixed logit model) and predicted the vaccine uptake probability for a hypothetical vaccine at different prices in each province to provide insights into real-world vaccination. The attributes of the hypothetical vaccine were chosen to closely resemble the Sinovac-CoronaVac vaccine selected from the vaccine attribute levels in the questionnaire [36]. All models were conducted using simulated maximum likelihood in the Apollo package (version 0.2.8) for R (version 4.2.0) [37].

## Results

### Sample characteristics

A total of 12,193 participants were recruited, and 12,000 participants were included in our analysis, resulting in 144,000 choices with each participant answering twelve choice tasks.

Among these participants, only 132 (1.10%) participants refused all COVID-19 vaccines regardless of vaccine characteristics. In total, 51.1% participants were male, 38.6% were 18–39 years old, and 60.5% were from urban areas, which represented a similar demographic distribution to the 2019 nationwide census [31]. Nearly half (45.8%) of all participants worked in a public setting, and all participants were covered by public health insurance, with 70.6% enrolled in UEBMI (Table 2).

## Main effects of vaccine attribute preferences

In the main results from the extended mixed logit model (Table 3), we found that adults in China generally preferred to have a vaccine compared with opt-out (coefficient 2.882, 95% Confidence Interval [2.809, 2.954]). The positive estimate of the alternative-specific constant for the first option (coefficient 0.121, [0.097, 0.145]) indicated that participants tended to choose the first option, in line with the known left-to-right reading bias from the literature [35]. Our results also revealed that participants had a higher preference for vaccines with lower risk (coefficient of high risk -1.158, [-1.359, -0.958]), longer duration of protection (coefficient 1.272, [1.016, 1.529]), higher efficacy (coefficient 1.063, [0.840, 1.287]), injection over oral administration (coefficient of oral vaccine -0.211, [-0.377, -0.046]), and fewer doses (coefficient of 2 doses -0.205, [-0.233, -0.176]). In addition, an imported vaccine was associated with a decrease in vaccine utility (coefficient -0.807, [-0.846, -0.767]).

## Heterogeneity among adults' vaccine preferences

There was significant heterogeneity in vaccine attribute preferences by individual characteristics including age group, education level, marital status, work situation, income level, chronic conditions, and vaccination history (Fig 1). In particular, we found that highly educated adults showed higher sensitivity to negative vaccine attributes (i.e., attributes with negative means of the preference parameters $v_m$ for non-reference levels; such attributes included price, risk, being imported, several doses). Participants with college-level education and above were more sensitive to vaccine attributes of price (coefficient -0.312, [-0.370, -0.253]) and imported vaccine (coefficient -0.941, [-1.186, -0.697]) than those who did not exceed primary school-level education. In addition, participants who were vaccinated, married, or were older were less sensitive to high risk and price. On the other hand, regarding positive vaccine attributes, participants with high income displayed higher sensitivity to long protection duration and high efficacy while participants diagnosed with a chronic disease were less sensitive to these attributes, compared with the reference group. Detailed results for the interaction effects from the extended mixed logit model are provided in S3 Table.

## Vaccine uptake probabilities

The predicted uptake probability for the baseline vaccine (free, no risk of severe side-effects, protection duration of 6 months, 50% vaccine effectiveness, one dose-injected, and domestically produced) was 95.3%. In Fig 2, we present the change in vaccine uptake probabilities by changing one vaccine attribute at a time compared with the baseline vaccine. The vaccine uptake probability increased by 2.7%, when vaccine protection duration increased from 6 months to life long. The vaccine uptake probability decreased by 12.1% when the domestic vaccine was replaced with an imported vaccine and decreased by 19.7% when vaccine price changed from free to 600 RMB.

In Fig 3, we depict the effect of price on the predicted uptake probabilities for the baseline vaccine. As price increased from free (baseline) to 1,000 RMB, the predicted vaccine uptake probability dropped from 95.3% to 63.1%. We also observed steepest decrease in the predicted

**Table 2. Sample descriptive characteristics.**

| Characteristics | Participants, n (%) | Population of China, % |
|---|---|---|
| **Sex** | | |
| Male (Reference) | 6,127 (51.1) | 51.1 |
| Female | 5,873 (48.9) | 48.9 |
| **Age Group** | | |
| 18–39 years (Reference) | 4,629 (38.6) | 42.5 |
| 40–59 years | 4,659 (38.8) | 43.9 |
| 60–70 years | 2,712 (22.6) | 14.6 |
| **Education** | | |
| Primary school or less (Reference) | 1,281 (10.7) | 30.4 |
| Middle and high school | 6,228 (51.9) | 55.0 |
| College and above | 4,491 (37.4) | 14.6 |
| **Marriage** | | |
| Unmarried (Reference) | 8,702 (72.5) | 74.0 |
| Married | 3,298 (27.5) | 26.0 |
| **Current Residence (Province)** | | |
| Anhui | 400 (3.3) | 4.5 |
| Beijing | 480 (4.0) | 1.5 |
| Chongqing | 360 (3.0) | 2.2 |
| Fujian | 360 (3.0) | 2.8 |
| Gansu | 360 (3.0) | 1.9 |
| Guangdong | 400 (3.3) | 8.2 |
| Guangxi | 400 (3.3) | 3.5 |
| Guizhou | 360 (3.0) | 2.6 |
| Hainan | 360 (3.0) | 0.7 |
| Hebei | 400 (3.3) | 5.4 |
| Heilongjiang | 360 (3.0) | 2.7 |
| Henan | 400 (3.3) | 6.9 |
| Hubei | 400 (3.3) | 4.2 |
| Hunan | 400 (3.3) | 4.9 |
| Jiangsu | 400 (3.3) | 5.8 |
| Jiangxi | 400 (3.3) | 3.3 |
| Jilin | 360 (3.0) | 1.9 |
| Liaoning | 400 (3.3) | 3.1 |
| Nei Mongol | 360 (3.0) | 1.8 |
| Ningxia | 360 (3.0) | 0.5 |
| Qinghai | 360 (3.0) | 0.4 |
| Shaanxi | 360 (3.0) | 2.8 |
| Shandong | 400 (3.3) | 7.2 |
| Shanghai | 480 (4.0) | 1.7 |
| Shanxi | 360 (3.0) | 2.7 |
| Sichuan | 400 (3.3) | 6.0 |
| Tianjin | 400 (3.3) | 1.1 |
| Tibet | 360 (3.0) | 0.3 |
| Xinjiang | 360 (3.0) | 1.8 |
| Yunnan | 400 (3.3) | 3.5 |
| Zhejiang | 400 (3.3) | 4.2 |
| **Residence** | | |

(*Continued*)

**Table 2.** (Continued)

| Characteristics | Participants, n (%) | Population of China, % |
|---|---|---|
| Rural (Reference) | 4,744 (39.5) | 39.4 |
| Urban | 7,256 (60.5) | 60.6 |
| **Work situations** | | |
| Work in a private setting (including freelancers) (Reference) | 3,484 (29.0) | - |
| Work in a public setting | 5,491 (45.8) | - |
| No job (including students, unemployment, retirement, and other reasons) | 3,025 (25.2) | - |
| **Insurance type*** | | |
| BMIURR (Reference) | 3,531 (29.4) | 24.5 |
| UEBMI | 8,469 (70.6) | 72.4 |
| No insurance | 0 (0.0) | 3.1 |
| **Total household income (RMB)** | | |
| < 60,000 (Reference) | 1,936 (16.1) | - |
| 60,000–149,999 | 6,687 (55.7) | - |
| ≥ 150,000 | 3,377 (28.1) | - |
| **Chronic conditions** | | |
| Not ever (Reference) | 11,085 (92.4) | - |
| Ever diagnosed with a chronic disease | 915 (7.6) | - |
| **COVID-19 vaccination history** | | |
| Not yet (Reference) | 4,840 (40.3) | - |
| Has been vaccinated at least one dose | 7,160 (59.7) | - |

*Abbreviations: BMIURR, Basic Medical Insurance for Urban and Rural Residents; UEBMI, Urban Employees Basic Medical Insurance.

uptake probability from 88.9% to 80.6% when the vaccine price increased from 400 to 600 RMB. When stratified by sex, age, and education populations (S1–S3 Figs), we found that female, older, and middle-level educated people were less sensitive to price increase, whereas male, middle-aged, and highly educated people were more sensitive, which is in line with the findings from the S3 Table.

### Geographical heterogeneity across price levels in vaccine preferences

To explore geographical heterogeneity in vaccine uptake probability, we estimated vaccine uptake probabilities for the hypothetical Sinovac-CoronaVac vaccine (moderate risk, protection duration of 6 months, 50% vaccine effectiveness, two dose-injected, and domestically produced) at different prices for each province in China (Fig 4). As the price of the vaccine increased, heterogeneity in predicted vaccine uptake probabilities across provinces gradually widened. When the vaccine was free, the highest vaccine uptake probability by province ranged from 76.5% (Shanghai) to 89.7% (Jiangxi)—this range widened from 49.7% (Shanghai) to 71.5% (Heilongjiang), when the price of the vaccine increased to 600 RMB. The provinces with relatively low vaccine uptake probabilities, such as Shanghai, Liaoning, Jiangsu, Sichuan, and Chongqing, are primarily concentrated in coastal and central areas of the country.

### Discussion

Utilizing a discrete choice experiment and mixed logit models, this study examined vaccine preferences with regard to vaccine characteristics and quantification of how the intent to receive vaccination changes with different COVID-19 vaccine attributes [9, 29]. We found a

**Table 3. Main effects from the extended mixed logit model.**

| Attribute and attribute levels | Coefficient ($v_m$)[a] | 95% CI of $v_m$ | SD ($\sigma_m$)[b] | 95% CI of $\sigma_m$ |
|---|---|---|---|---|
| ASC | 2.882 | [2.809, 2.954] | - | - |
| ASC (1st) | 0.121 | [0.097, 0.145] | - | - |
| Price (per 100 RMB) | -0.224 | [-0.275, -0.173] | 0.371 | [0.359, 0.383] |
| Risk | | | | |
| No risk | Ref | - | | |
| Moderate risk | -0.844 | [-1.031, -0.658] | 0.694 | [0.639, 0.749] |
| High risk | -1.158 | [-1.359, -0.958] | 0.758 | [0.708, 0.808] |
| Duration of protection | | | | |
| 6 months | Ref | - | - | - |
| 12 months | 0.393 | [0.209, 0.578] | -0.361 | [-0.488, -0.233] |
| Life long | 1.272 | [1.016, 1.529] | 1.412 | [1.363, 1.462] |
| Degree of efficacy | | | | |
| 50% | Ref | - | - | - |
| 70% | 0.402 | [0.207, 0.598] | -0.439 | [-0.574, -0.304] |
| 90% | 1.063 | [0.840, 1.287] | -1.182 | [-1.228, -1.136] |
| Oral vaccine | -0.211 | [-0.377, -0.046] | -0.718 | [-0.778, -0.659] |
| Frequency of vaccination | | | | |
| 1 dose | Ref | - | - | - |
| 2 doses | -0.205 | [-0.233, -0.176] | 0.031 | [-0.093, 0.155] |
| 3 doses | -0.148 | [-0.296, 0.000] | 0.623 | [0.574, 0.673] |
| Imported vaccine | -0.653 | [-0.864, -0.443] | 1.427 | [1.377, 1.477] |
| Number of observations | 144,000 | | | |
| Number of participants | 12,000 | | | |
| Wald χ2/LRχ2 | 114,802.2 | | | |
| P>χ2 | <0.001 | | | |
| Akaike Information Criterion (AIC) | 201,926.1 | | | |
| Bayesian information criteria (BIC) | 203,546.1 | | | |
| Log pseudo-likelihood | -100,799.1 | | | |

[a] The signs of $v_m$ indicated whether the reference group preferred the vaccine attribute or not; positive coefficient meant this attribute was preferred and otherwise meant this attribute was not preferred. See Methods Section for further details of variable notations and model specifications.

[b] The relationship of 95% CI of $\sigma_m$ and 0 indicated whether there existed heterogeneity in this vaccine attribute level. If included, then no significant heterogeneity was found and if not, significant heterogeneity was found. Negative estimate of $\sigma_m$ originates from the statistical software (Apollo package for R) where the sign of $\sigma_m$ is not constrained for ease of parameter estimation [38].

Abbreviations: CI, confidence interval; SD, standard deviation; ASC, alternative-specific constant.

widespread openness for COVID-19 vaccines in China, in which only 132 out of 12,000 (1.1%) participants opted for no vaccine in all twelve tasks, regardless of vaccine characteristics. Vaccine preference increased with longer protection duration and higher efficacy, and lower risks of side effects. Findings on heterogeneities in vaccine preferences also elucidated different uptake probabilities across geographic regions, education levels, age groups, and other subpopulations in relation to vaccine price. Furthermore, this study contributes to the existing vaccine preference literature by employing a large nationwide sample and using a wide range of vaccine attributes.

We found an exceptionally high percentage of people in China who hold a positive attitude toward preventing and controlling COVID-19 through vaccination. Only 1.1% of participants opted for no vaccine in our study. Our findings show a sharp increase in the percentage of

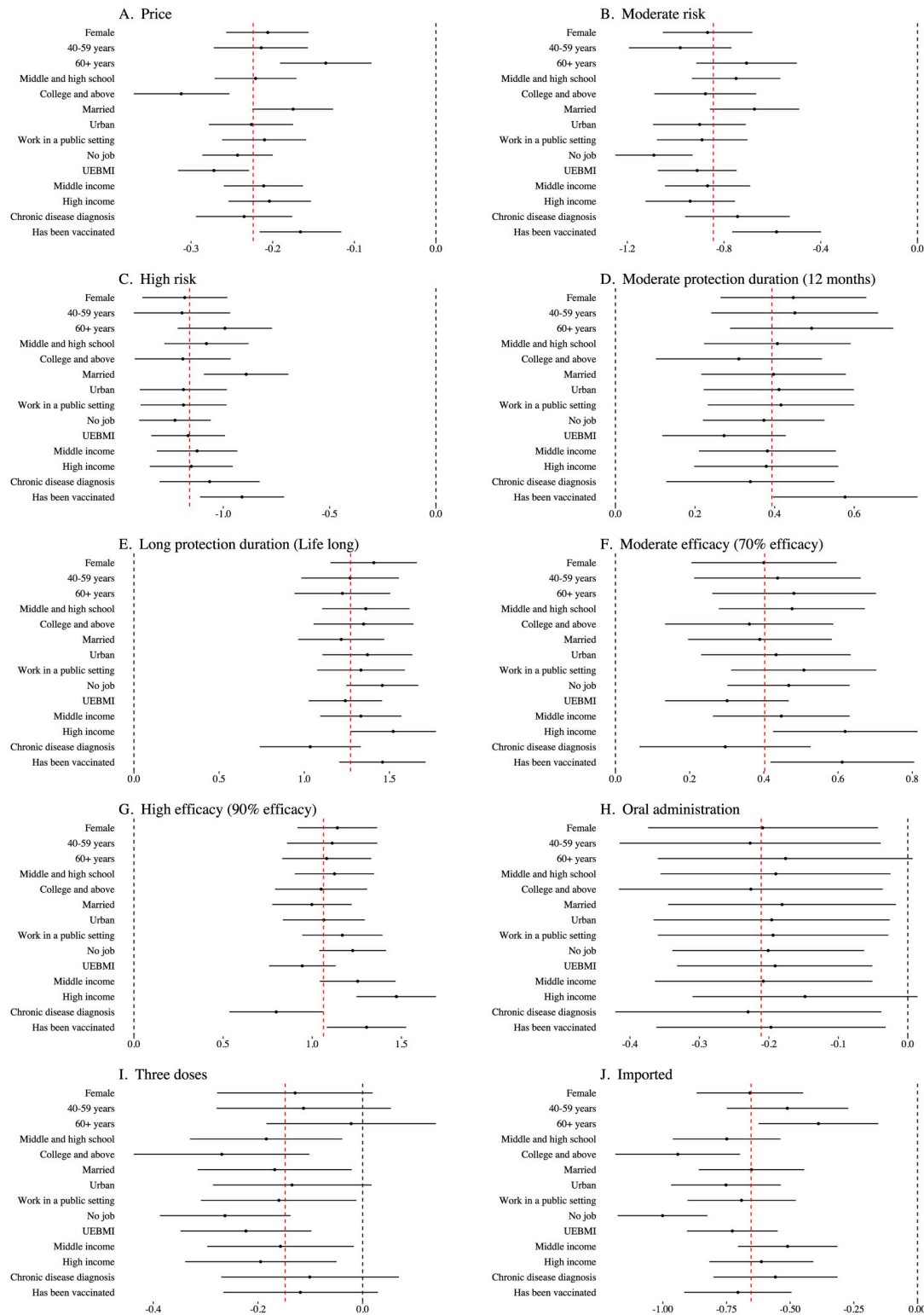

**Fig 1. Main effects plus interaction effects across groups for different vaccine attributes estimated from the extended mixed logit model.** Notes: Each subfigure presents the main effect of vaccine attribute $m$ plus interaction effects with individual's characteristics (i.e., $v_m + \gamma_{mk}$, see Methods Section for further details of variable notations and model specifications.). The red dashed line indicated the mean of main effect of the reference group (male, 18–39 years, primary school or less, unmarried, rural, work in a private setting, UEBMI, < 60,000 RMB, not ever diagnosed with a chronic disease, not

vaccinated by COVID-19 vaccines). Heterogneity is considered significant if confidence interval of each individual's characteistic does not intersect with the red dashed line (detailed reports are provided in S3 Table). The black dashed line represented the point of no preference. The relationship of the confidence interval to the black dashed line indicated whether this attribute was preferred or not preferred by this group. The interactions between two doses and participants' characteristics were excluded in the extended mixed logit model.

people who hold positive preferences toward vaccination, compared with the findings from an earlier study in China prior to vaccine availability that showed 80% of Chinese respondents would be willing to take the vaccine when it becomes available [18]. In addition, we found a high predicted vaccine uptake probability of 95.3% for the baseline vaccine, which had attributes including free costs, no risk of severe side-effects, six months duration of protection, 50% vaccine efficacy, one-dose injection, and domestic production. An earlier study in China found a predicted uptake probability of 84.8% under conditions including 85.0% vaccine effectiveness, 1/100,000 severe side-effect risks, free costs, one dose, delivery at tertiary vaccination sites (i.e., county hospitals and above), two years duration of protection and 90.0% of acquaintances vaccinated [16]. The higher predicted uptake probability in our study suggests that

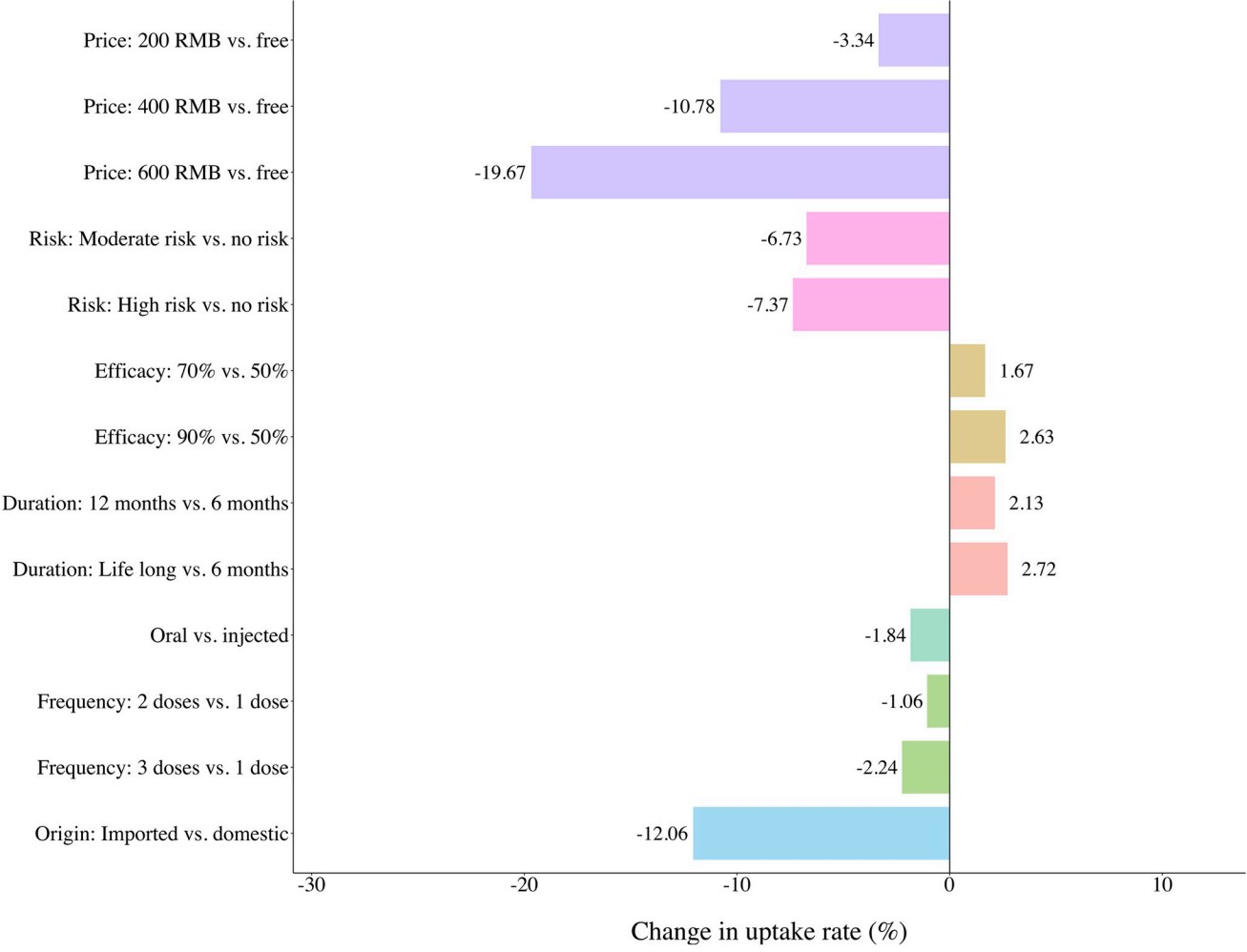

**Fig 2. Effects of changing the attribute levels on uptake probabilities from baseline vaccine uptake of 95.3%.**

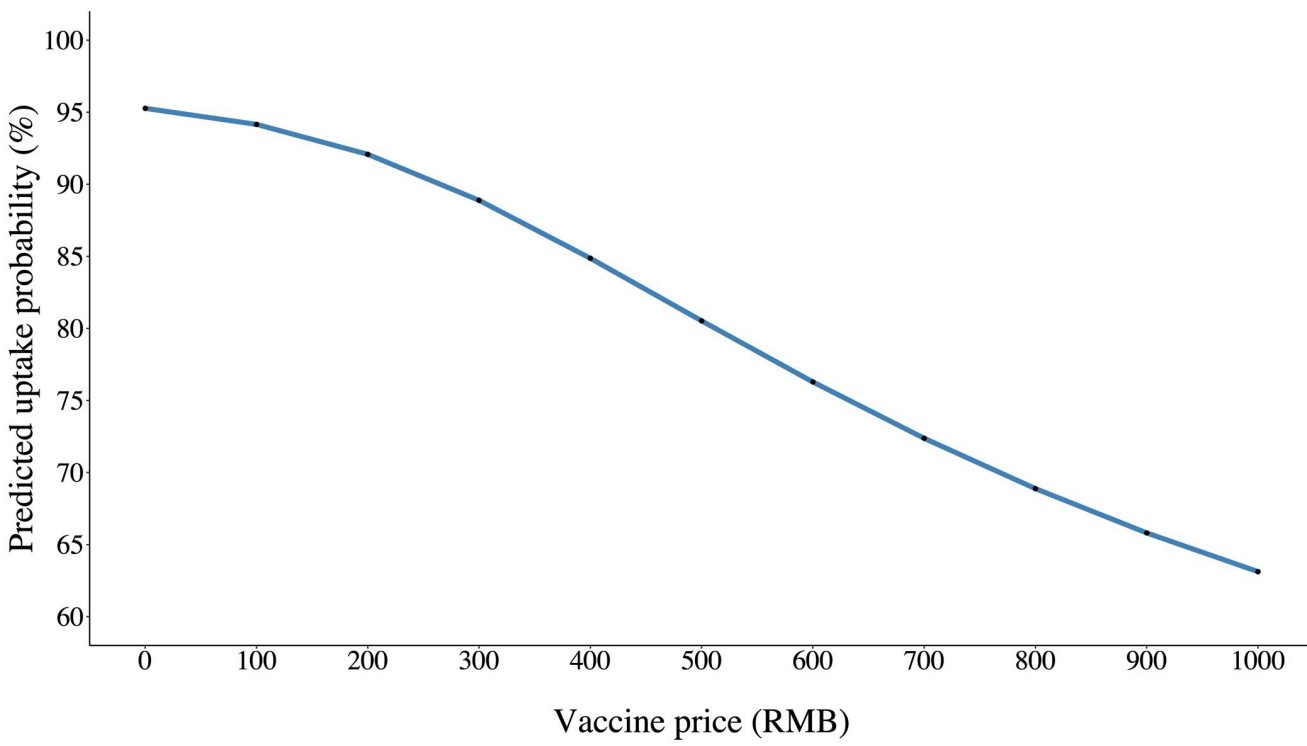

**Fig 3. Effect of price on predicted uptake probabilities from the extended mixed effect model.**

vaccine promotion and increased knowledge of the COVID-19 pandemic over time contributed to increased vaccine uptake [40].

Our findings also demonstrate the impact of vaccine attributes on vaccine preferences. For example, whether the vaccine was imported or domestically manufactured significantly impacted participants' uptake probabilities, with the vaccine uptake probability decreasing by 12.1% when the domestic vaccine was replaced with an imported vaccine. This finding is in line with most existing literature, such as a cross-country analysis conducted in Latin American [41] and a DCE conducted in India [20], where studies identified a higher preference for domestic vaccines rather than imported vaccines [4, 42–44]. However, our findings contradict a prior study in China which found higher preference for imported vaccines instead of domestically manufactured vaccines using a smaller sample [18]. The inconsistency in findings relating to whether there was higher preference for imported or domestic vaccines may result from constraints in sample size and representativeness, as the prior study that found a preference for imported vaccines did not have a large sample size and was not nationally representative. Considering the larger sample size and national representativeness of this study as well as the high levels of trust placed in the government and healthcare system in China [45], we have sufficient evidence to conclude that there is a higher preference for domestic vaccines rather than imported vaccines in China.

Vaccine price is also an important attribute that showed a significant impact from our analysis. The uptake probability of 95.3% dropped sharply to 63.1% when the free baseline vaccine increased in price to 1,000 RMB (US $154). While this was a steep decrease in vaccine uptake probability, the results still indicate that more than half the people would be willing to receive vaccinations if the price was 1,000 RMB. This preference contradicts previous studies in early

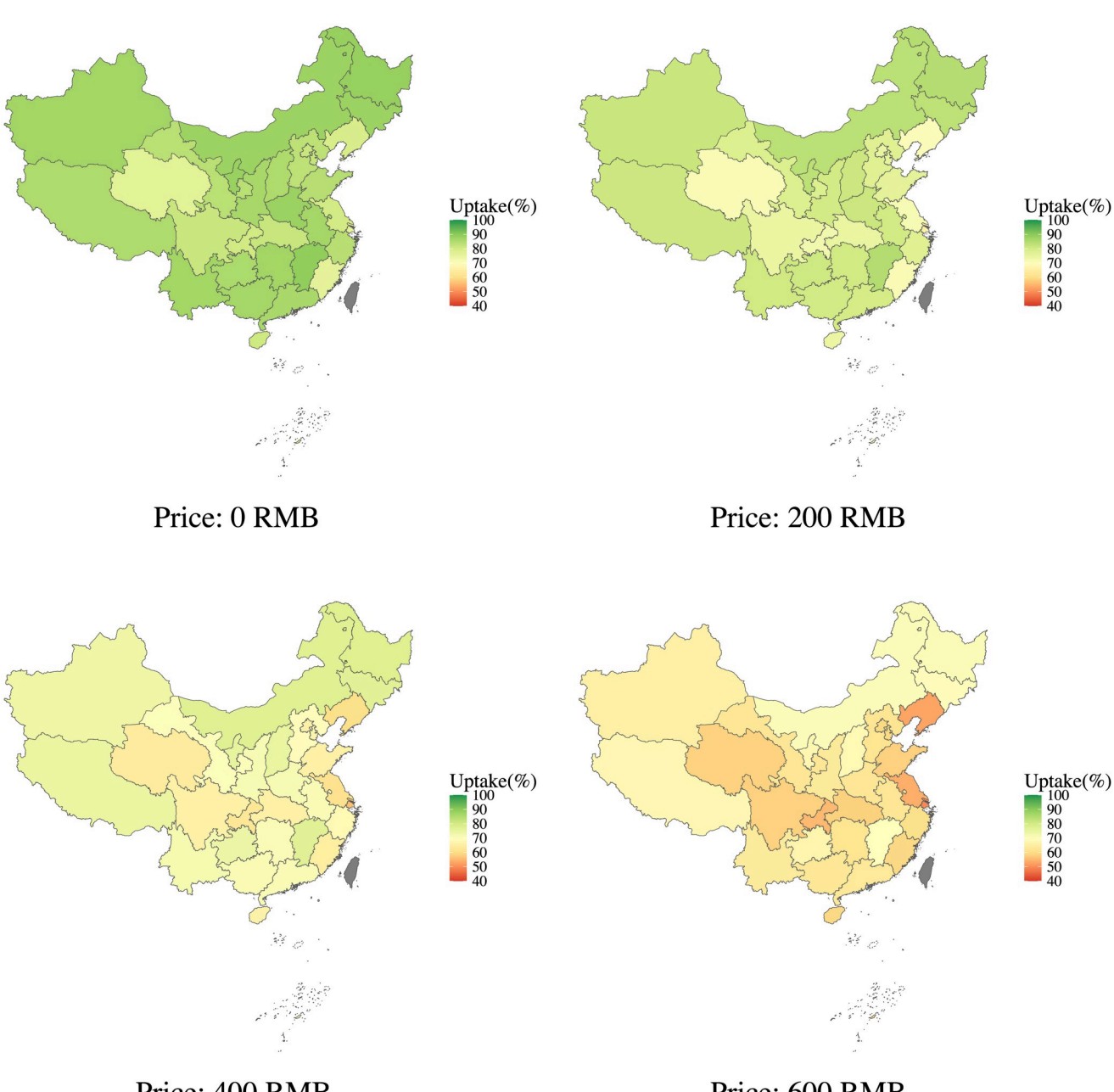

**Fig 4. Predicted vaccine uptake probabilities across provinces from the mixed effect model at different prices.** Note: Areas in grey color represent missing data. Base layer data source: GaryBikini/ChinaAdminDivisonSHP: ChinaAdminDivisonSHP v24.02.06 (zenodo.org) Republished from [39] under a CC BY license, with permission from GaryBikini, original copyright 2020.

2020 that revealed a willingness-to-pay (WTP) within the range of 100–300 RMB for COVID-19 vaccines in China [46, 47], where WTP refers to the price when the predicted vaccine uptake probability is 50%. The contradiction may be explained by vaccination promotion efforts. When the majority of people considered the price to be too high when the expected Chinese vaccine price was about 1,000 RMB in August 2020 [48], numerous strategies were adopted in China to facilitate vaccine uptake [49], some of which may have had a positive

impact on WTP, such as health communication campaigns [50]. In September 2020, the Chinese public already favored a vaccine price of 600 RMB [51]. It is likely that by 2021, the emphasis on vaccination had become stronger such that WTP increased to 1,000 RMB, which supports our findings. Another possible explanation for the increase in WTP from 600 RMB to 1000 RMB may be that by the time the participants were surveyed in this study, the Chinese government had already announced that COVID-19 vaccinations would be offered free to Chinese citizens [52] and so participants may have believed that they would not have to pay for COVID-19 vaccines out-of-pocket regardless of vaccine price. Knowing that vaccines are provided for free might change the behavior of participants through cognitive mechanisms such as anchoring effects, which is the tendency for participants to rely heavily on the first piece of information when making decisions [53]. When vaccines are announced as free, this sets an initial anchor of zero cost, which may lead participants to express a high WTP as a form of positive feedback or approval of the free vaccine initiative, even if they do not actually expect to pay that amount–this may also explain our findings of the lower sensitivity to vaccine price among adults with high income. However, not all participants may have been aware of the free COVID-19 vaccination announcement and there could have been uncertainty about how long the free COVID-19 vaccination may last for. Furthermore, this anchoring effect may be moderated by the fact that influenza vaccines in the past were typically paid out-of-pocket and were only partially or fully subsidized for children and older adults via health insurance or local government funds in economically developed regions in China [54, 55]. As the pandemic progressed and the Chinese government relaxed COVID-19 restrictions, the perceived severity of COVID-19 decreased among the public, and the lessened threat of being infected with COVID-19 was likely accompanied by a decrease in WTP. In fact, WTP of COVID-19 booster dose in China was estimated to decrease to 149 RMB from a 2022 survey [56].

This study also sheds light on heterogeneities regarding vaccine preferences. We demonstrated geographical heterogeneities in predicted vaccine uptake probability between Chinese provinces as vaccine price changed. In particular, when vaccine price increased to 600 RMB, predicted vaccine uptake probabilities dropped to about 40.0% - 50.0% for coastal regions such as Shanghai and Liaoning. This could be attributed to the higher sensitivity to price changes in coastal regions as a result of higher living costs despite potentially higher average incomes [57]. With regards to heterogeneities in vaccine preferences across education levels, we found that as the vaccine price increased, the predicted vaccine uptake probabilities among those who received college education decreased substantially relative to those who received primary school or no school education. The observed relationship between education level and vaccine preference is supported by several previous studies [58–61]. One possible reason is that people with higher education levels may have greater access to information, which increases the likelihood of being exposed to misinformation about COVID-19 vaccines, leading to vaccine hesitancy [62]. People who endorse more misinformation are less likely to follow behaviors to prevent COVID-19 [63]. Specifically, social media usage was identified as the main source of COVID-19 vaccine misinformation in China, and public health authorities and professionals should make an active effort to debunk false claims [64]. However, it is important to note that education level is found to be positively associated with vaccine uptake in most countries and regions [65, 66]. The discrepancy in findings may be attributed to the varying level of health education or vaccine-related education in relation to general education level in different countries (i.e., if a higher general education level necessarily leads to higher awareness and knowledge of the benefits of vaccination). Understanding the varying degrees of vaccine preferences among each region and subpopulations with different education levels as well as the underlying reasons can support local policymakers to mitigate these factors and increase the likelihood of successfully executing vaccination campaigns.

The findings of this study have several policy implications. First, we found that price is a critical factor in determining vaccine uptake probability. Currently, COVID-19 vaccination is free of charge in China, but the findings from this study indicate that policymakers must seriously consider the level at which vaccine prices are set, when vaccines are no longer free in the future. Furthermore, targeted vaccine campaigns in specific geographic regions can be helpful in promoting vaccine uptake, especially in regions that are more price-sensitive. Second, while there has been mixed evidence in the literature regarding preferences between imported vaccines and domestic vaccines, our nationally representative findings from a large sample size confirm higher preferences for domestic vaccines. However, levels of trust in the government are situational and may change in the future. Therefore, it is important to facilitate strong and sustainable trust in the government and healthcare system by increasing credibility and promoting open communication [45]. In particular, sending consistent and unified vaccine-related messages on media is much needed to facilitate trust and combat misinformation [63, 65, 67]. Third, our results indicate that oral administration is not beneficial and vaccines should require as few doses as possible.

This study also has several limitations. First, participants were selected from a pool of registered adults with KuRunData who agreed to participate in the survey, and thus the study may be subject to selection bias depending on profiles of platform users. In particular, it presented challenges in reaching older individuals, and thus our sample may not fully represent the entire adult population, especially due to the absence of participants aged over 70. To minimize bias, we utilized quota sampling and stratified sampling by age, gender, and residence to increase representativeness of the general adult population. Second, we included only a selected set of vaccine attributes, leaving out several other attributes that have also been discussed in vaccine literature such as vaccination site [16] and distance to vaccination site [68], to ensure that participants had the cognitive capacity to consider all attributes when making choices. To guarantee that the most important attributes were included, we conducted a systematic qualitative investigation and literature review beforehand to identify the most important attributes and specify the corresponding levels for our study. Third, it is crucial to acknowledge the common problem of cognitive difficulty present in discrete choice experiments, which is aggravated when DCEs are online without support, and the possible discrepancy between participants' stated preferred choices and actual choice made in reality.

## Conclusion

Our findings suggest that to facilitate higher adults' preferences for COVID-19 vaccines, future vaccines should be domestically produced, offer longer protection period, have higher efficacy, and be of no risk to health. We provide empirical evidence on general adults' vaccine preferences with relevance for policymakers in China, as well as other countries with similar concerns, to inform future best practices with respect to pandemic preparedness. Our results also offer recommendations for adjusting vaccine subsidy policies and vaccine design policies for targeted vaccine campaigns in general to facilitate wider vaccination coverage.

## Supporting information

**S1 Checklist. Inclusivity in global research.**
(DOCX)

**S1 Text. Questionnaire in English and Chinese.**
(DOCX)

**S2 Text. Survey experiment development.**
(DOCX)

**S1 Fig. Effect of price on predicted uptake probabilities by sex from the extended mixed effect model.**
(TIFF)

**S2 Fig. Effect of price on predicted uptake probabilities by age from the extended mixed effect model.**
(TIFF)

**S3 Fig. Effect of price on predicted uptake probabilities by education level from the extended mixed effect model.**
(TIFF)

**S1 Table. Example of choice task.**
(DOCX)

**S2 Table. Results of the basic mixed logit model.**
(DOCX)

**S3 Table. Detailed results of the extended mixed logit model.**
(DOCX)

## Author Contributions

**Conceptualization:** Jonas Wachinger, Juntao Yang, Till Bärnighausen, Simiao Chen.

**Data curation:** Fengyun Yu, Qun Wang, Zhong Cao, Jonas Wachinger, Simiao Chen.

**Formal analysis:** Fengyun Yu, Lirui Jiao, Zhong Cao.

**Funding acquisition:** Juntao Yang, Simiao Chen.

**Investigation:** Fengyun Yu, Manuela De Allegri, Simiao Chen.

**Methodology:** Fengyun Yu, Qun Wang, Jonas Wachinger.

**Project administration:** Qun Wang.

**Resources:** Zhong Cao, Wenjin Chen, Chao Wang, Juntao Yang, Simiao Chen.

**Supervision:** Qiushi Chen, Qun Wang, Pascal Geldsetzer, Juntao Yang, Lan Xue, Till Bärnighausen.

**Validation:** Manuela De Allegri.

**Visualization:** Qiushi Chen.

**Writing – original draft:** Fengyun Yu, Lirui Jiao, Qiushi Chen, Xuedi Ma.

**Writing – review & editing:** Fengyun Yu, Lirui Jiao, Qiushi Chen, Manuela De Allegri, Jonas Wachinger, Zhangfeng Jin, Aditi Bunker, Pascal Geldsetzer, Lan Xue, Till Bärnighausen, Simiao Chen.

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
