## [Decision Letter · Decision Letter 0]

28 Feb 2024

PGPH-D-23-02386

Preferences regarding COVID-19 vaccination among 12,000 adults in China: a cross-sectional discrete choice experiment

Dear Dr. Chen,

Thank you for submitting your manuscript to PLOS Global Public Health. After careful consideration, we feel that it has merit but does not fully meet PLOS Global Public Health’s publication criteria as it currently stands. Therefore, we invite you to submit a revised version of the manuscript that addresses the points raised during the review process.

The manuscript has been evaluated by two reviewers, and their comments are available below. The reviewers have raised a number of concerns that need attention. They request additional information on methodological aspects of the study and revisions to the discussion. We note that one of the reviewers has recommended that you cite specific previously published works. As always, we recommend that you please review and evaluate the requested works to determine whether they are relevant and should be cited. It is not a requirement to cite these works. We appreciate your attention to this request.

We look forward to receiving your revised manuscript.

Kind regards,

Marianne Clemence

Staff Editor

Journal Requirements:

If you did not receive any funding for this study, please simply state: “The authors received no specific funding for this work.

3. Please provide separate figure files in .tif or .eps format only and remove any figures embedded in your manuscript file. Please also ensure all files are under our size limit of 10MB.

4. Some material included in your submission may be copyrighted. According to PLOS’s copyright policy, authors who use figures or other material (e.g., graphics, clipart, maps) from another author or copyright holder must demonstrate or obtain permission to publish this material under the Creative Commons Attribution 4.0 International (CC BY 4.0) License used by PLOS journals. Please closely review the details of PLOS’s copyright requirements here: PLOS Licenses and Copyright. If you need to request permissions from a copyright holder, you may use PLOS's Copyright Content Permission form.

Potential Copyright Issues:

Figs 4: please (a) provide a direct link to the base layer of the map (i.e., the country or region border shape) and ensure this is also included in the figure legend; and (b) provide a link to the terms of use / license information for the base layer image or shapefile. We cannot publish proprietary or copyrighted maps (e.g. Google Maps, Mapquest) and the terms of use for your map base layer must be compatible with our CC-BY 4.0 license. 

"

Additional Editor Comments (if provided):

Reviewers' comments:

Reviewer's Responses to Questions

**Comments to the Author**

1. Does this manuscript meet PLOS Global Public Health’s publication criteria? Is the manuscript technically sound, and do the data support the conclusions? The manuscript must describe methodologically and ethically rigorous research with conclusions that are appropriately drawn based on the data presented.

Reviewer #1: Yes

Reviewer #2: Partly

2. Has the statistical analysis been performed appropriately and rigorously?

Reviewer #1: Yes

Reviewer #2: Yes

3. Have the authors made all data underlying the findings in their manuscript fully available (please refer to the Data Availability Statement at the start of the manuscript PDF file)?

Reviewer #1: Yes

Reviewer #2: Yes

4. Is the manuscript presented in an intelligible fashion and written in standard English?

Reviewer #1: Yes

Reviewer #2: Yes

5. Review Comments to the Author

Reviewer #1: I thank the authors for conducting a very thorough study to explore the impact of vaccine attributes and other determinants on vaccine uptake. To my knowledge, no other study that has looked at so many different attributes influencing COVID-19 vaccine uptake.

This study, using the discrete choice experiment, determined various vaccine related attributes that influence vaccine uptake. The authors looked at many different attributes, including vaccine efficacy, dose requirement, country of production and potential side effects. They also explored individual level determinants that are associated with vaccine uptake such as age, education level, work situation, income level, etc. Moreover, the authors looked at vaccine preferences among populations living in different geographical locations in China. And finally, they predicted vaccine uptake probability of a hypothetical vaccine to further demonstrate the geographical heterogeneities based on pricing in vaccine uptake. I agree with the authors that these findings would help in designing targeted vaccine campaigns for vaccines, especially adult vaccines, in the future.

The authors of this study have tried to create a nationally representative sample while acknowledging the limitations of selection bias.

Overall, the study is well written. However, I would request the authors to re-assess their discussion section to make it more thorough and ensure all the major findings are discussed (especially the points that are mentioned in the abstract).

Specifically, I have a few minor comments for the authors.

Comments:

Introduction:

1. Please mention the date (month and year) instead of “At the time of writing,..”. With the ever-changing situation related to COVID-19, it will allow the reader to get a general idea of what the situation of the pandemic was when this study was written.

Discussion:

1. Line 398-402: The authors compare their findings of vaccine uptake preferences in July 2021, after vaccines were available and had been administered for over six-months (China began vaccinating in December 2020), with studies from other countries (France and Chile) that predicted vaccine uptake when vaccines were not available. For a fair comparison, please share stats from studies from China, if any, that were conducted before the vaccines were available. Trust in the government, communication strategies, etc. can be very different in different countries.

2. Line 431-450: The authors suggest that health communication efforts in China could have had a positive impact on people’s willingness to pay for the vaccine. For some people, that might be the case. However, for most of the population, the ability to pay for the vaccine, especially when the cost is as steep as 1,000 RMB would be the deciding factor. Therefore, the authors should further elaborate on what could be the reason for this finding. Cost of the vaccine is a big barrier when it comes to predicting vaccine uptake. Is it possible that the survey participants believed that whatever the price of the vaccine, they will not have to pay for it out of pocket and it will be covered by their insurance? How does the health insurance work in China? Would insurance have covered the cost of the vaccine for those who had health insurance?

Interestingly, in this study, adults with high income were not significantly sensitive to the price of the vaccine. (Figure 1 a)

3. Lines 452-460: Vaccine uptake probability across different provinces was high (76.5% to 89.7%) when the price was 0 RMB, which reduced to 49.8% to 71.5% when the price was increased to 600 RMB. (line 373-376) However, the authors quote that a contributing factor to this drop could be the different composition of occupational groups in different states. This does not appear to be a plausible explanation unless a significant proportion of population is different based on occupation. Could there be other reasons such as differences on the bases of urban and rural, education, affordability, income etc. It’s not clear that if price changes, why would the healthcare workers suddenly be less willing to be vaccinated in Shanghai.

4. Can the authors also elaborate on why the health care workers are less willing to get vaccinated?

5. Line 462-470: The study found that education was inversely associated with vaccine uptake. This is contradictory to findings from other studies (Viswanath et al. BMC Public Health (2021) 21:818 https://doi.org/10.1186/s12889-021-10862-1) that found education to be positively associated with vaccine uptake. Moreover, according to the world bank (https://www.worldbank.org/en/country/china/overview#:~:text=China%20is%20now%20an%20upper,in%20upper%2Dmiddle%20income%20countries), China is now an upper-middle-income country. So, comparing the findings of this study with LMICs is not justified.

6. Misinformation has been a huge menace to the COVID-19 pandemic. Study by Dhawan et al., (Dhawan et al. (2021) COVID-19 News and Misinformation: Do They Matter for Public Health Prevention?, Journal of Health Communication, 26:11, 799-808, DOI: 10.1080/10810730.2021.2010841) found that people who endorse more misinformation are less likely to follow behaviors to prevent COVID-19. Since the authors mention misinformation as one of the plausible reasons behind low vaccine uptake among people with higher education, can the authors also talk about the issue of misinformation in China? It will be of interest to the readers to learn about the spread of misinformation in China affecting vaccine uptake despite having a more regulated media environment.

7. The authors can refer to studies below to provide recommendations for targeted vaccine campaigns to promote vaccine uptake based on the findings of this study:

a. Viswanath et al. (2021) Individual and social determinants of COVID-19 vaccine uptake. BMC Public Health 21, 818. https://doi.org/10.1186/s12889-021-10862-1

b. Dhawan et al. (2021) COVID-19 News and Misinformation: Do They Matter for Public Healt

---

## [Decision Letter · Decision Letter 1]

3 Jun 2024

Preferences regarding COVID-19 vaccination among 12,000 adults in China: a cross-sectional discrete choice experiment

PGPH-D-23-02386R1

Dear Dr Chen,

We are pleased to inform you that your manuscript 'Preferences regarding COVID-19 vaccination among 12,000 adults in China: a cross-sectional discrete choice experiment' has been provisionally accepted for publication in PLOS Global Public Health.

Best regards,

Julia Robinson

Executive Editor

Reviewer Comments (if any, and for reference):

Reviewer's Responses to Questions

**Comments to the Author**

1. If the authors have adequately addressed your comments raised in a previous round of review and you feel that this manuscript is now acceptable for publication, you may indicate that here to bypass the “Comments to the Author” section, enter your conflict of interest statement in the “Confidential to Editor” section, and submit your "Accept" recommendation.

Reviewer #2: All comments have been addressed

2. Does this manuscript meet PLOS Global Public Health’s publication criteria? Is the manuscript technically sound, and do the data support the conclusions? The manuscript must describe methodologically and ethically rigorous research with conclusions that are appropriately drawn based on the data presented.

Reviewer #2: Yes

3. Has the statistical analysis been performed appropriately and rigorously?

Reviewer #2: Yes

4. Have the authors made all data underlying the findings in their manuscript fully available (please refer to the Data Availability Statement at the start of the manuscript PDF file)?

Reviewer #2: Yes

5. Is the manuscript presented in an intelligible fashion and written in standard English?

Reviewer #2: Yes

6. Review Comments to the Author

Reviewer #2: Thanks the authors for the responses. All my concerns have been addressed.

7. PLOS authors have the option to publish the peer review history of their article (what does this mean?). If published, this will include your full peer review and any attached files.

**Do you want your identity to be public for this peer review?** For information about this choice, including consent withdrawal, please see our Privacy Policy.

Reviewer #2: No
